# Do cancer detection rates differ between transperineal and transrectal micro-ultrasound mpMRI-fusion-targeted prostate biopsies? A propensity score-matched study

**Arnas Rakauskas**[1]*, **Max Peters**[2], **Paul Martel**[1], **Peter S. N. van Rossum**[2], **Stefano La Rosa**[3,4], **Jean-Yves Meuwly**[5], **Beat Roth**[1], **Massimo Valerio**[1]

**1** Department of Urology, Lausanne University Hospital, Lausanne, Switzerland, **2** Department of Radiotherapy, University Medical Center Utrecht, Utrecht, The Netherlands, **3** Institute of Pathology, Lausanne University Hospital, Lausanne, Switzerland, **4** Department of Medicine and Surgery, Pathology Unit, University of Insubria, Varese, Italy, **5** Department of Radiology, Lausanne University Hospital, Lausanne, Switzerland

\* Rakauskas.arn@gmail.com

## Abstract

### Introduction

High-resolution micro-ultrasound (micro-US) is a novel precise imaging modality that allows targeted prostate biopsies and multiparametric magnet resonance imaging (mpMRI) fusion. Its high resolution relying on a 29 MHz transducer allows real-time visualisation of prostate cancer lesions; this might overcome the inaccuracy of conventional MRI-US fusion biopsy strategies. We compared cancer detection rates in patients who underwent transrectal (TR-B) versus transperineal (TP-B) MR-micro-US fusion biopsy.

### Materials and methods

1:2 propensity score matching was performed in 322 consecutive procedures: 56 TR-B and 266 TP-B. All prostate biopsies were performed using ExactVu™ micro-US system with mpMRI image fusion. Clinically significant disease was defined as grade group $\geq$2. The primary objective was to evaluate the detection of clinically significant disease according to access route. The secondary outcomes were to compare the respective detection rates of random and targeted biopsies stratified per access route and to evaluate micro-US for its potential added value.

### Results

47 men undergoing TR-B and 88 undergoing TP-B were matched for age, PSA, clinical stage, prostate volume, PIRADS score, number of mpMRI-visible lesions and indication to biopsy. The detection rates of clinically significant and of any prostate cancer did not differ between the two groups (45% TR-B vs 42% TP-B; p = 0.8, and 57% TR-B vs 59% TP-B; p = 0.9, respectively). Detection rates also did not differ significantly between random (p = 0.4) and targeted biopsies (p = 0.7) stratified per access route. Micro-US targeted biopsy detected 36 MRI-invisible lesions in 33 patients; 19% of these lesions were positive for

**Data Availability Statement:** All relevant data are within the manuscript and its Supporting Information files.

**Funding:** The author(s) received no specific funding for this work.

**Competing interests:** The authors have declared that no competing interests exist.

clinically significant disease. Overall, micro-US targeted biopsies upgraded 2% of patients to clinically significant disease that would have been missed otherwise.

## Conclusions

MR-micro-US-fusion TR-B and TP-B have similar diagnostic yields in terms of detection rates of clinically significant prostate cancer. Micro-US targeted biopsy appears to have an additional diagnostic value over systematic and MRI-targeted biopsies.

## 1. Introduction

The adoption of multiparametric magnetic resonance imaging (mpMRI) in the evaluation of patients for prostate cancer has enhanced our ability to detect clinically significant disease. Robust evidence from level 1 diagnostic studies shows that performing a mpMRI prior to prostate biopsy allows detection of more men with clinically significant disease than systematic biopsy with no prior imaging [1–3]. Consequently, current guidelines recommend performing mpMRI prior to any prostate biopsy; however, the ability to perform MRI-targeted biopsy is limited by the pitfalls of current MRI-US fusion strategies. Indeed, conventional transrectal ultrasound technology is unable to characterise prostatic tissue and to differentiate between benign and non-benign areas. This may lead to incorrect sampling due to needle deviation, fusion error, and/ or lack of operator experience. In addition, mpMRI can miss clinically significant lesions in specific settings [4]. Further optimization of the current biopsy strategy would require the adoption of a novel imaging modality that is able to characterise prostatic tissue, enable fusion with mpMRI, and perform fusion biopsy under direct control.

High-resolution micro-ultrasound (micro-US; ExactVu$^{TM}$) is a new imaging modality that enables the visualization and targeting of suspicious areas of the prostate in real time [5]. Micro-US operates at 29 MHz and provides a direct replacement for conventional ultrasound with a threefold improvement in spatial resolution. Micro-US devices use FusionVu$^{TM}$ mpMRI-micro-US image fusion software. A few recent studies suggest that micro-US-mpMRI image fusion may further increase the cancer detection rate versus conventional US-MRI fusion [6, 7].

Current micro-US hardware can perform transrectal biopsy (TR-B) and transperineal biopsy (TP-B) of the prostate (Figs 1 and 2). TP-B has been gaining popularity due to its higher cancer detection rate in the antero-apical zones, lower sepsis rate, and lower risk of rectal bleeding [8–10]. No study has yet assessed the impact of the access route on detection rates using a micro-US-mpMRI fusion device. We report our initial experience detecting clinically significant cancer with micro-US-mpMRI image fusion targeted TP-B versus TR-B and evaluate whether micro-US targeted biopsy confers additional diagnostic value.

## 2. Methods

### 2.1. Study population

This is a retrospective analysis of consecutive patients undergoing micro-US-mpMRI fusion biopsy between May 2018 and March 2020. All patients signed a consent form. This study is Research Ethics Committee-approved (Cantonal Commission on Ethics in Human Research (CER-VD), study number 2020–00396). Biopsies were offered to patients with elevated PSA levels and/or suspicious digital rectal examination and/or positive prostate mpMRI with lesions scored 3–5 according to the Prostate Imaging–Reporting and Data System (PI-RADS)

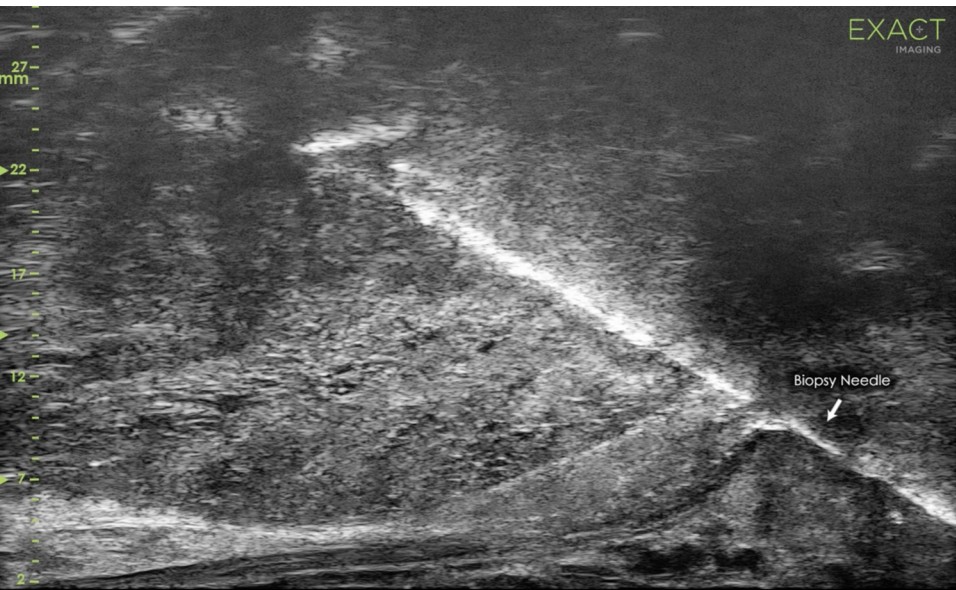

**Fig 1. Image of transrectal micro-US-targeted biopsy with no mpMRI fusion.**

v2.0 [11]. Patients with no mpMRI visible lesions were also included. In these cases, biopsies were mainly performed due to high PSA density (>0.15 ng/mL/cm3) and/or clinical suspicion of prostate cancer. Patients undergoing biopsy within an active surveillance protocol or for risk stratification were also included in the study.

## 2.2. Magnetic resonance protocol

All patients underwent a mpMRI prior to the biopsy. In our institution, the mpMRI protocol is standardized. Each 3T prostate mpMRI was acquired and interpreted according to PI-RADS

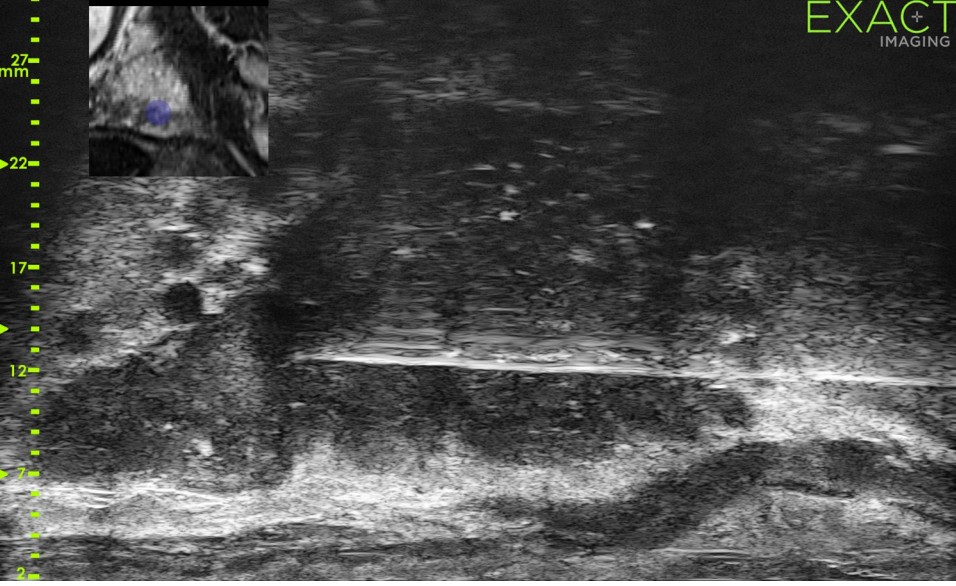

**Fig 2. Image of transperineal micro-US-mpMRI-targeted biopsy.**

and reported by a single expert radiologist. An endorectal coil was systematically employed; the multiparametric protocol included T1-weighted, T2-weighted, diffusion-weighted imaging with corresponding apparent diffusion coefficient map, and dynamic contrast enhanced sequences. Prior to biopsy, all mpMRI were reviewed in a dedicated uroradiology meeting where a single expert radiologist with over 15 years' experience in prostate MRI (JYM) annotated the relevant lesions. Delineated T2-weighed and apparent diffusion coefficient sequences were uploaded into the ExactVu system (Exact Imaging™, Markham, Canada).

## 2.3 Biopsy protocol

All prostate biopsies were performed using the ExactVu system. A single expert urologist performed or supervised every procedure. Two approaches were offered: TR-B or TP-B. The indications to each approach were discussed with every patient taking into consideration the risk of infection, the anaesthesia implications, and the location of the lesions. TP-B was generally preferred over TR-B for anterior lesions and for defining focal therapy eligibility [12]. TR-B was performed with the patient in a left lateral position under local anaesthesia and oral fluoroquinolone prophylaxis over three days. TP-B was performed with the patient in the lithotomy position under sedation, general or local anesthesia, and single intravenous 2 g ceftriaxone injection.

The biopsy protocol was the same regardless of the approach. For TR-B, the probe was moved free-hand, whereas for TP-B the probe was mounted on a stepper (D&K technologies GmbH, Barum, Germany). In both approaches, the prostate gland was scanned from right to left with an imaging depth of 5cm, followed by an imaging depth of 3cm to visualise the fine prostate architecture. The gland was inspected to identify MRI-detected lesions and additional lesions apparent on micro-US. The latter lesions were graded according to the PRI-MUS score, which was specifically developed for interpreting micro-US imaging [13]. In case of MRI-visible lesions, the micro-US and MRI images were aligned using the FusionVu™ rigid fusion software embedded in the ExactVu device. Micro-US-MRI-targeted biopsies with 2–4 cores per lesion were performed first, followed by MRI-invisible micro-US-positive lesion targeted biopsies, and finally by 10–12 cores standard systematic biopsies. All biopsies were performed using a conventional spring-loaded gun and a trocar sharpened needle deployed through a specific single-use clip-on guide for transrectal or transperineal biopsies manufactured by ExactVu. Specimens were separately potted and analysed by an expert genitourinary pathologist according to the ISUP recommendation [14].

## 2.4 Primary outcome

We evaluated whether there was a difference in the detection rates of clinically significant disease according to the access route (TP-B vs TR-B) in men undergoing micro-US-MRI-fusion targeted and systematic biopsy. Clinically significant prostate cancer was defined as any prognostic grade group ≥2 disease at histology.

## 2.5 Secondary outcomes

Our secondary outcomes were the comparison of clinically significant prostate cancer detection rates between random and targeted biopsies stratified per access route (TP-B vs TR-B) and evaluation of the added value of micro-US in detecting clinically significant prostate cancer missed by MRI-targeted and systematic biopsy. The added value of micro-US targeting was defined as the number of men who were upgraded to clinically significant prostate cancer based on micro-US targeted biopsy only.

## 2.6 Statistical methods

Patients who underwent any prostate cancer treatment prior to the biopsy or had less than 10 biopsy cores obtained were excluded from further analysis. A 1:2 propensity score match pairing was first performed to match TR-B and TP-B groups. Patients were matched using the following variables: age, PSA, clinical stage, prostate volume, mpMRI score (PI-RADS 1–2 vs 3–5), number of mpMRI-visible lesions and indication to biopsy. Nearest neighbour matching was used without replacement and a calliper width of 0.20 of the standard deviation of the logit of the propensity score was adopted as maximum difference for matched cases [15].

Continuous variables were reported with medians (interquartile ranges [IQR]) or means (standard deviation [SD]) when appropriate, and categorical variables with frequencies and proportions. Differences in continuous and categorical variables were tested using the Mann-Whitney U test or unpaired T-test and $\chi^2$ or Fisher's exact test as appropriate.

Before the propensity score was created and subsequent analyses performed, single imputation was used to correct for missing data, which was assumed to be missing at random.

R version 3.5.3 was used for all statistical analyses (R Foundation for Statistical Computing, Vienna, Austria). The statistical significance was set at a p-value <0.05.

## 3. Results

The initial database comprised 322 prostate biopsies. After 1:2 (TR-B: TP-B) propensity score match pairing, 47 TR-B and 88 TP-B patients were included in the final analysis. Of these 135 patients, 64% were biopsy-naïve, 9% had previous negative and 27% had previous positive biopsies. Median age was 66 years (IQR, 59–72 yrs) and median PSA was 7.2 ng/ml (5.4–10.7 ng/ml). Most patients (81%) presented with nonpalpable disease; mpMRI was positive in 60% patients with one or more PIRADS 3–5 lesions (Table 1). Variables were well balanced between the TR-B and TP-B groups after matching (standardized mean differences all below 0.1). Complete histological findings stratified by access route and type of biopsy are presented in Table 2 and displayed per indication to biopsy in Fig 3.

**Table 1. Patients' characteristics.**

| Variable | Overall matched population | TR-B | TP-B | p value |
|---|---|---|---|---|
| N (%) | 135 | 47 (35%) | 88 (65%) | |
| Age, median (IQR) | 66 (59–72) | 66 (60–70) | 67 (59–72) | 0.96 |
| PSA baseline, median (IQR) | 7.2 (5.4–10.7) | 7 (4.8–9.5) | 7.7 (5.4–13.3) | 0.86 |
| Clinical tumor stage (%) | | | | 0.45 |
| T1c | 109 (81%) | 37 (79%) | 72 (82%) | |
| cT2 | 24 (18%) | 10 (21%) | 14 (16%) | |
| cT3 | 2 (1%) | 0 | 2 (2%) | |
| PIRADS score (%) | | | | 0.8 |
| 1–2 | 54 (40%) | 20 (43%) | 34 (39%) | |
| 3–5 | 81 (60%) | 27 (57%) | 54 (61%) | |
| Prostate volume, median (IQR) | 42 (32–62.5) | 40 (30–61) | 45 (32–63) | 0.41 |
| Number of mpMRI suspicious lesions, median, range | 1 (0–3) | 1 (0–2) | 1 (0–3) | 0.69 |
| Previous biopsy status (%) | | | | 0.5 |
| Biopsy naïve | 87 (64%) | 31 (66%) | 56 (64%) | |
| Previous negative | 12 (9%) | 4 (9%) | 8 (9%) | |
| Previous positive | 36 (27%) | 12 (26%) | 24 (27%) | |

**Table 2. Histology results stratified by access route and type of biopsy.**

| Biopsy type | TRANSRECTAL | | | TRANSPERINEAL | | |
|---|---|---|---|---|---|---|
| | SYSTEMATIC | TARGETED | OVERALL | SYSTEMATIC | TARGETED | OVERALL |
| No biopsy taken, median (IQR) | 12 (8–13) | 3 (0–6) | 14 (12–16) | 15 (10–22) | 3 (0–6) | 17 (14–24) |
| No positive cores, median (IQR) | 1 (0–2) | 1 (0–3) | 1 (0–5) | 0 (0–2) | 1 (0–3) | 2 (0–5) |
| MCCL, median (IQR) | 3 (2–7) | 6 (1–13) | 6 (2–8) | 4 (2–6) | 4 (0–9) | 6 (3–9) |
| Grade group, cases (%) | 1–13 (54%) | 1–4 (24%) | 1–10 (37%) | 1–16 (46%) | 1–10 (27%) | 1–17 (33%) |
| | 2–9 (38%) | 2–9 (53%) | 2–14 (52%) | 2–11 (32%) | 2–16 (43%) | 2–21 (40%) |
| | 3–1 (4%) | 3–1 (6%) | 3–0 | 3–4 (11%) | 3–6 (16%) | 3–8 (15%) |
| | 4–1 (4%) | 4–2 (12%) | 4–2 (7%) | 4–4 (11%) | 4–3 (8%) | 4–4 (8%) |
| | 5–0 | 5–1 (6%) | 5–1 (4%) | 5–0 | 5–2 (5%) | 5–2 (4%) |

### 3.1. Primary outcome

There was no statistically significant difference in detection rates of clinically significant cancer between TP-B and TR-B, nor in the detection rate of any prostate cancer (42% vs 45%, p = 0.8, and 59% TP-B vs 57% TR-B, p = 0.9, respectively).

### 3.2. Secondary outcomes

The median (IQR) number of positive cores was not significantly different between systematic TR-B and TP-B (0, [0–2] vs 1 [0–3], p = 0.2) and targeted TR-B and TP-B (1 [0–3] vs 1 [0–3], p = 0.5). The mean (SD) maximum cancer core length (MCCL) was also not significantly different between systematic TR-B and TP-B (4.0mm [2.7mm] vs 4.3mm [3.1mm], p = 0.7) and targeted TR-B and TP-B (6.9mm [5.1mm] vs 5.5mm [6.2mm], p = 0.4). There was no difference either in the detection rate of clinically significant prostate cancer between systematic

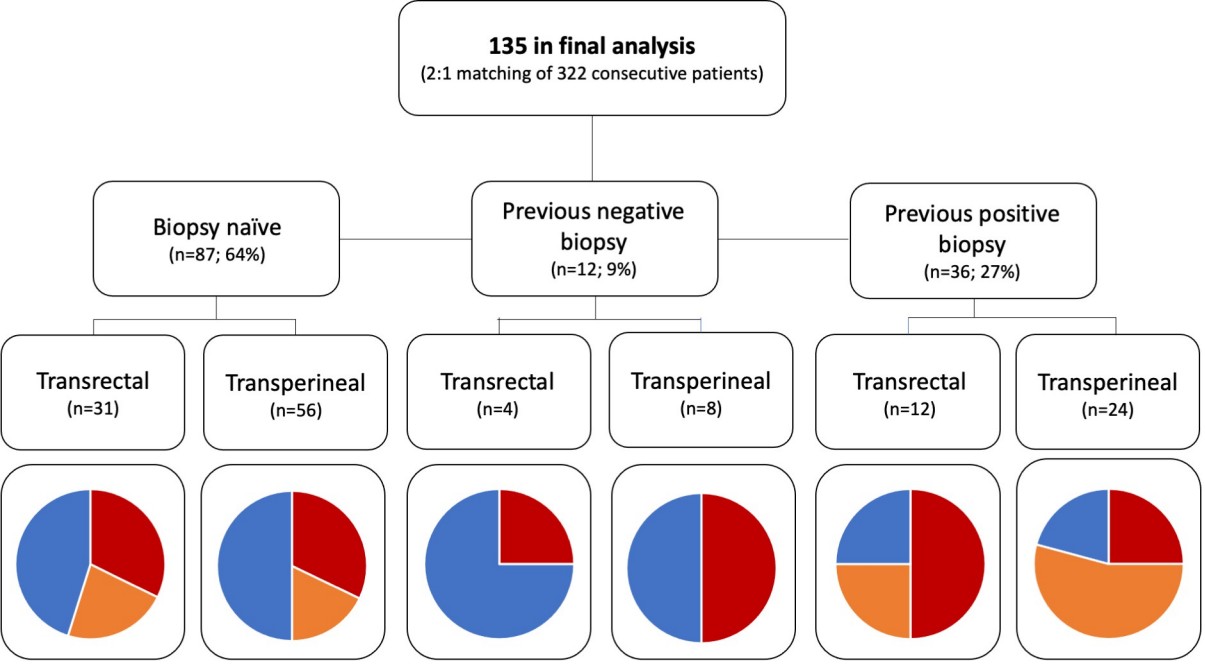

**Fig 3. Cancer detection rates per indication and biopsy approach (red = clinically significant cancer, orange = clinically insignificant cancer, blue—no cancer).**

**Table 3. Systematic, MRI, and micro-US-targeted biopsy findings in the 12 men with additional micro-US lesions invisible at MRI.** X = biopsy type not performed as the PIRADS score was 1–2 in these patients.

|  | Prognostic Grade Group Systematic biopsy | Prognostic Grade Group MRI-targeted biopsy | Prognostic Grade Group Micro-US-targeted biopsy |
|---|---|---|---|
| 1. | 1 | x | 1 |
| 2. | 2 | x | 2 |
| 3. | 1 | x | 1 |
| 4. | 4 | 4 | 4 |
| 5. | 1 | x | 2 |
| 6. | 2 | 0 | 2 |
| 7. | 0 | x | 2 |
| 8. | 2 | x | 2 |
| 9. | 1 | x | 1 |
| 10. | 1 | 2 | 1 |
| 11. | 1 | 2 | 2 |
| 12. | 1 | 3 | 1 |

TR-B or TP-B (n = 19, 43.2% vs n = 35, 41.2%, p = 0.9) and targeted TR-B and TP-B (n = 14, 41.2% vs n = 30, 47.6%, p = 0.7) on a patient level. Targeted biopsies detected more clinically significant disease and less clinically insignificant disease in both approaches (TR-P $p < 0.01$; TP-B p = 0.03) on a patient level."

In the 135 patients, micro-US detected 36 MRI invisible lesions in 33 patients (24%) which were sampled with a median of 2 (2–3) cores per lesion. In 7 out of 36 lesions (19%), clinically significant cancer was detected; in 5 out of 36 lesions (14%), clinically insignificant cancer was detected. No cancer was detected in 24 out of 36 lesions (67%). The median MCCL of micro-US-positive targeted biopsy was 2.5 mm (2–3.5mm). Upgrading to clinically significant disease based on micro-US targeting occurred in only two patients, which corresponds to an added value at a patient level of 2% (Table 3).

## 4. Discussion

This study shows that there is no difference between TR-B and TR-P in terms of detection rate of clinically significant and insignificant prostate cancer using micro-US mpMRI fusion-targeted biopsy. In one out of four patients, micro-US detected additional MRI invisible lesions; of these, 19% were positive for clinically significant disease. In our series, this amounts to a 2% upgrading to clinically significant disease based on micro-US targeted biopsy.

This is the first study comparing the two employing micro-US-mpMRI fusion imaging. Systematic reviews [10, 16] show similar cancer detection rates between TR-B and TP-B using conventional ultrasound. However, most of the studies included in these quantitative analyses did not routinely perform mpMRI and targeted biopsies. Therefore, their findings cannot be entirely applied to current clinical practice. In a prospective study by Pepe et al., 200 men underwent saturation biopsy followed by a targeted TR-B and a TP-B in case of PIRADS 4–5 lesion [9]. In that unprecedented study, a selection bias was elegantly overcome by submitting every patient to a saturation biopsy plus 4 core mpMRI-targeted TR-B plus cognitive-targeted TP-B. The detection rate for significant disease was lower in their TR-B group (66.7%) than in their TP-B group (93.3%). For anterior lesions, the cancer detection rate was significantly higher with the transperineal than the transrectal approach (93.7% vs 25%, p = 0.0001). In our study, we did not compare the anterior lesion cancer detection rate between the two approaches as the presence of an anterior lesion was per se an indication to TP-B. In another

prospective study by Ber et al. comprising 77 patients, mpMRI-targeted TR-B and TP-B were performed during the same intervention in case of PIRADS ≥3 lesion [17]. Thirty-two participants (31%) were diagnosed with clinically significant disease, defined as grade group ≥2 and/ or cancer-core-length >/ = 6 mm. The absolute difference for significant disease detection in the per-protocol analysis was 15.6 (CI 90% 3.2–27.9%) in favor of TP-B (p = 0.029). In our study, high and comparable cancer detection rates between the two approaches might be due to the precision of micro-US mpMRI-fusion-targeted biopsies. Indeed, the high-definition of micro-US allows real-time targeting and might mitigate the sampling error related to needle deviation and the incorrect fusion, sometimes observed with the transrectal approach. Also, the overall accuracy of TR-B in our series might be higher than in other series as only peripheral zone lesions were included in our analysis for the reasons mentioned above.

Our study also evaluated the added value of micro-US to MRI and systematic biopsies. While a recent systematic review involving 1125 patients did not identify an additional value for micro-US targeting compared to purely mpMRI targeting in terms of prostate biopsy cancer detection rate [18], in our study we found an added value of micro-US in 2% of patients resulting in a potential change in management. These results are in line with a prospective study by Lughezzani et al. comparing the detection rates of clinically significant disease between mpMRI and micro-US targeted biopsies [6]. In their study, the authors reported that micro-US-targeted biopsy identified an additional 3% of men with clinically significant disease that would have been missed by mpMRI. Another prospective multi-center study of 1040 patients mirrors our results with a 4% upgrading derived from micro-US-targeted lesions invisible on pre-biopsy MRI [7]. While an added diagnostic value of 2% to 4% in detecting clinically significant disease in the three aforementioned studies seems to be relatively small, we believe that the high overall cancer detection rate in our study as well as in the available literature is a direct result of the accuracy of micro-US, which enables accurate definition of prostatic anatomy and might reduce cognitive or software errors observed with other MRI fusion strategies.

The OPTIMUM trial by Klotz et al. will evaluate whether micro-US alone could be an alternative to mpMRI and conventional ultrasound fusion [19]. This three-arm randomized control trial will compare clinically significant cancer detection rates between micro-US only, micro-US-mpMRI fusion, and conventional US-mpMRI-targeted fusion biopsies. Health economic analysis will also be performed as a secondary outcome. The results of the OPTIMUM trial will further clarify whether micro-US should be regarded as a replacement or an add-on test to systematic and mpMRI-targeted biopsy.

Our study has some limitations. First, it's a retrospective and a hypothesis generating study that is prone to selection bias. We tried to minimize this potential bias by using a strict matched-pair strategy. Second, our patients with anterior lesions systematically underwent TP-B; therefore, we could not compare the clinically significant disease detection rates in anterior zones between the two approaches. Finally, our study is limited by its heterogeneity in terms of indications to biopsy and the relatively small sample size. Awaiting more robust evidence from studies designed to compare micro-US against standard of care in a more homogenous and larger cohort [19], the present results may foster interest in this novel technology that appears to be equally effective and useful for both a transrectal and a transperineal approach.

## 5. Conclusion

TR-B appears to be as precise as TP-B, at least in men with no visible anterior lesions. Our study confirms that micro-US-targeted biopsy provides an additional diagnostic value in

detecting clinically significant disease to the standard of care provided by MRI-targeted and systematic biopsy. Whether micro-US can be a replacement for or an add-on test to standard of care is being further explored in ongoing studies.

## Supporting information

**S1 Dataset.**
(XLSX)

## Author Contributions

**Conceptualization:** Max Peters.

**Data curation:** Arnas Rakauskas, Max Peters.

**Formal analysis:** Max Peters.

**Investigation:** Paul Martel, Peter S. N. van Rossum, Stefano La Rosa, Jean-Yves Meuwly, Beat Roth, Massimo Valerio.

**Methodology:** Arnas Rakauskas, Max Peters, Peter S. N. van Rossum, Massimo Valerio.

**Project administration:** Massimo Valerio.

**Resources:** Arnas Rakauskas, Paul Martel, Stefano La Rosa, Jean-Yves Meuwly, Beat Roth.

**Software:** Max Peters, Peter S. N. van Rossum.

**Supervision:** Beat Roth, Massimo Valerio.

**Validation:** Jean-Yves Meuwly, Beat Roth, Massimo Valerio.

**Writing – original draft:** Arnas Rakauskas, Massimo Valerio.

**Writing – review & editing:** Max Peters, Paul Martel, Peter S. N. van Rossum, Stefano La Rosa, Jean-Yves Meuwly, Beat Roth, Massimo Valerio.

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
