## [Decision Letter · Decision Letter 0]

22 Sep 2022

PONE-D-22-12037Do cancer detection rates differ between transperineal and transrectal micro-ultrasound mpMRI-fusion-targeted prostate biopsies? A propensity score-matched studyPLOS ONE

Dear Dr. Rakauskas,

Thank you for submitting your manuscript to PLOS ONE. After careful consideration, we feel that it has merit but does not fully meet PLOS ONE’s publication criteria as it currently stands. Therefore, we invite you to submit a revised version of the manuscript that addresses the points raised during the review process.

Please revise. 

We look forward to receiving your revised manuscript.

Kind regards,

Academic Editor

PLOS ONE

Reviewers' comments:

Reviewer's Responses to Questions

**Comments to the Author**

1. Is the manuscript technically sound, and do the data support the conclusions?

Reviewer #1: Partly

Reviewer #2: Yes

2. Has the statistical analysis been performed appropriately and rigorously? 

Reviewer #1: Yes

Reviewer #2: Yes

3. Have the authors made all data underlying the findings in their manuscript fully available?

Reviewer #1: No

Reviewer #2: Yes

4. Is the manuscript presented in an intelligible fashion and written in standard English?

Reviewer #1: Yes

Reviewer #2: Yes

5. Review Comments to the Author

Reviewer #1: The authors retrospectively compared the overall prostate cancer (PCa) and clinically significant prostate cancer (csPCa) detection rate between transperineal (TP) and transrectal (TR) micro-ultrasound mpMRI-fusion-targeted prostate biopsies in a series of only 135 patients. The secondary outcomes were to compare the detection rates of random and targeted biopsies stratified per access route and to evaluate micro-US for its potential added value. They found that the PCa and csPCA detection rates between TR and TP were similar. Detection rates also did not differ between random and targeted biopsies stratified per access route TR vs. TP. Additionally; they reported that targeted micro-ultrasound biopsies upgraded 2% of patients to csPCa

The micro-ultrasound is a promisor new technique that may improve PCa diagnosis and therefore new data is welcome. The novelty of this study relies in the comparison of TP vs. TR route employing MRI and micro-ultrasound. The sample size is small and this is the main problem of this study - 47 vs. 88 cases make hard to anyone to conclude that both biopsy routes are similar. The authors recognize this issue and justify by saying that this paper represents their initial results. Based on that I recommend rejection until a larger sample is achievable by this group. The paper has also some methodology problems that require major revision if the editor consider acceptance of it.

Major revisions

1) If the main purpose is to compare the detection rate of TR versus TP biopsy; the exclusion of anterior zone tumors definitively compromise the results and probably is the main cause of no detection rate difference between the two groups. If I agree with the removal of anterior tumors at MRI because it would favor the TP group (author´s justification), I also should remove posterior tumors in accordance because it would favor the TR route.

2) The small sample size is the major limitation of this study. They reported on only 135 patients of which ~42-45% of cases were positive for csPCa. This small number of cases probably jeopardize the study’s results and the conclusions. For example, the csPCa detection rate between random and MRI guided were similar in this trial, which is against the whole body of the MRI guide prostate biopsy literature that shows that MRI guided fragments have a higher chance of csPCa when compared to random fragments. The small number of cases may explain this contradictory finding

3) The author say that this is their preliminary result but the last case was included two years ago. Once the sample is of only 135 cases, why the authors did not include the most recent performed cases?

4) The add value of micro-ultrasound was of only 2% but even so the authors came to the conclusion that it has an additional diagnostic value over systematic and MRI-targeted biopsies. I don´t think this 2% is clinically significant. For example, it is acknowledge that a biopsy can be avoided in patients with PIRADS-2, previous negative biopsy and a PSA density below 0.15 because in this situation we would miss only 4 to 5% of csPCa. If missing 5% of csPCa is acceptable I don´t think that an increase of only 2% by adding micro-ultrasound has any clinical benefits. According to the results, I would have to employ micro-ultrasound in 50 cases to diagnose an additional csPCa case. Once these are the initial results perhaps, the added micro-ultrasound value may be higher with the inclusion of more cases.

5) At methodology line 103, the author say that all MRI was reviewed which is important. However, it is not clear how this was done? How many radiologists participated at this “uroradiology meeting”? The uroradiologists had experience reading multiparametric MRI? If there is disagreement regarding the assigned PIRADS how this was solved?

6) Despite the propensity score-matched process the groups are still unbalanced. The transperineal group has a higher number of sampled fragments. The author think that this may have an impact at detection rate between the two groups?

7) At discussion and conclusion the author highlighted the potential benefits and accuracy of micro-ultrasound that are not based in their results. They argue that TR and TB detection rates were similar because the benefits of micro-ultrasound. However, both groups underwent biopsy with micro-ultrasound therefore what is been compared is not the advantage of micro-ultrasound but the difference of TR versus TP route. A third transrectal and a fourth transperineal arms without micro-ultrasound would be necessary to attest that the lack of difference between groups 1 and 2 were due the advantage of micro-ultrasound or its higher accuracy.

8) The micro-ultrasound detected 36 MRI invisible lesions in 33 patients but only 19% of them were CsPCa at biopsy result and the final additional benefit in increasing csPCa detection was of only 2%. Where is the micro-ultrasound advantage? 67% of suspicious lesions at micro-ultrasound had no cancer at all......

Minor revision

1) Why the table 1 that compare the patient´s characteristics between the two groups do not contain the p-values? The p-values must be included at tables and not only at the text.

2) I think table 3 is unnecessary

3) This study is a hypothesis generating study and this should be highlighted at study limitations at discussion section (lines 260 -267).

4) Line 119 change “TR-P” to “TP-B”

5) MRI was “positive” in 60% of cases (line 168) How many cases were micro-ultrasound positive I mean PRIMUS 3 to 5?

6) Why to mention the Future trial that did not compared TP versus TR biopsies ? Instead they compared Fusion vs. Cognitive vs. in Bore biopsy? The Future did not evaluate micro-ultrasound either. Furthermore it is well known that Future trial is an underpowered study because at sample size calculation of the trial they needed around 400 PIRADS 3-5 lesions but they had only 234 cases with suspicious MRI.

Reviewer #2: It is an interesting and present issue. The microultrasound have been shown promissed results and probably will have a role in prostate cancer diagnostic scenario. The authors report results from a retrospective matched groups comparison which showed no difference between detections rates between transperineal and transrectal prostate biopsies using micro-ultrassound. Despite the retrospective design study decreases its conclusions streght these results deserve to be published. The choice for a prostate biopsy route should take the detection rate in consideration. This study reinforce a tendency showed in previous publications which compared perineal and transrectal route using ordinary ultrasound devices.

In my opinion, some aspects could be clarified to improve this report:

1. Do the authors believe that the report of anterior-only positive biopsies on perineal route could be interesting for comparison between perineal and transrectal detection rates?

2. Regarding the low PPV value of PIRADS 3 MRI targeted biopsies, do the authors believe that a secondary analysis including only PIRADS 3 splited from PIRADS 4-5 could improve the results? How many PIRADS 3 lesions presented different characteristics on micro-US views, for example? Thinking in this issue, In Table 1 - Patients Characteristics, could the rate of PIRADS 3 be individualized?

3. Were the micro-US target biopsies performed by a blinded operator? Specifically, had the radiologists previous access to the MRI images, before biopsy? Do the authors believe that this fact could interfer in positive rates of micro-US target biopsies when compared to MRI fusion biopsies?

4. Do the rates of micro-US targeted biopsies differ from random biopsies? In my opinion, it is not clear in the report.

6. PLOS authors have the option to publish the peer review history of their article (what does this mean?). If published, this will include your full peer review and any attached files.

Reviewer #1: No

Reviewer #2: No

---

## [Author Response · Author response to Decision Letter 0]

13 Oct 2022

The response to reviewers is available in the separate word file as requested.

---

## [Decision Letter · Decision Letter 1]

26 Dec 2022

Do cancer detection rates differ between transperineal and transrectal micro-ultrasound mpMRI-fusion-targeted prostate biopsies? A propensity score-matched study

PONE-D-22-12037R1

Dear Dr. Rakauskas,

We’re pleased to inform you that your manuscript has been judged scientifically suitable for publication and will be formally accepted for publication once it meets all outstanding technical requirements.

Kind regards,

Academic Editor

PLOS ONE

Additional Editor Comments (optional):

Reviewers' comments:

Reviewer's Responses to Questions

**Comments to the Author**

1. If the authors have adequately addressed your comments raised in a previous round of review and you feel that this manuscript is now acceptable for publication, you may indicate that here to bypass the “Comments to the Author” section, enter your conflict of interest statement in the “Confidential to Editor” section, and submit your "Accept" recommendation.

Reviewer #3: All comments have been addressed

Reviewer #4: All comments have been addressed

2. Is the manuscript technically sound, and do the data support the conclusions?

Reviewer #3: Yes

Reviewer #4: Yes

3. Has the statistical analysis been performed appropriately and rigorously? 

Reviewer #3: Yes

Reviewer #4: N/A

4. Have the authors made all data underlying the findings in their manuscript fully available?

Reviewer #3: Yes

Reviewer #4: Yes

5. Is the manuscript presented in an intelligible fashion and written in standard English?

Reviewer #3: Yes

Reviewer #4: Yes

6. Review Comments to the Author

Reviewer #3: This is a well written manuscript of a retrospective study of prostate gland biopsy. It tried to highlight the benefits of prostate micro-US and if this should be regarded as a replacement or an add-on test to systematic and mpMRI-targeted biopsy.

The manuscript had satisfied the journal requirements.

However, the standardized protocol in the study environment where a single expert radiologists interprets 3T prostate mpMRI and PI-RADS can be modified to allow for between 2 to 3 expert radiologists to interpret these results. This will reduce/eliminate single observer bias.

Reviewer #4: I reviewed the manuscript, editors' comments and the responses. To me this is an honest report of their observation and will add to the current knowledge. If I was supposed to write this, I would have the overall detection part first and then compare the results of different approaches to the biopsy. However, the relevant reader of this paper will find their way to interpret the findings.

7. PLOS authors have the option to publish the peer review history of their article (what does this mean?). If published, this will include your full peer review and any attached files.

Reviewer #3: No

Reviewer #4: No

---

## [Editor Report · Acceptance letter]

6 Jan 2023

PONE-D-22-12037R1 

Do cancer detection rates differ between transperineal and transrectal micro-ultrasound mpMRI-fusion-targeted prostate biopsies? A propensity score-matched study 

Dear Dr. Rakauskas:

I'm pleased to inform you that your manuscript has been deemed suitable for publication in PLOS ONE. Congratulations! Your manuscript is now with our production department. 

Kind regards, 

on behalf of

Dr. Robert Jeenchen Chen 

Academic Editor

PLOS ONE